Subject Areas:
environmental science/ecology/behaviour

Keywords:
cooperation, bioeconomic models, evolutionary game theory, small-scale fishery, fisher behaviour, compliance

Author for correspondence:
Eric Zettermann Dias de Azevedo
e-mail: ericzt@hotmail.com

# Risk tolerance and control perception in a game-theoretic bioeconomic model for small-scale fisheries

Eric Zettermann Dias de Azevedo[1,2],
David Valença Dantas[1] and Fábio G. Daura-Jorge[2]

[1]Departamento de Engenharia de Pesca e Ciências Biológicas, Universidado do Estado de Santa Catarina, Laguna, Brazil
[2]Programa de Pós-graduação em Ecologia, Departamento de Ecologia e Zoologia, Universidade Federal de Santa Catarina, Florianópolis, Santa Catarina, Brazil

  EZDA, 0000-0003-2399-7571; DVD, 0000-0001-6890-1313;
FGD-J, 0000-0003-2923-1446

Cooperation is generally the most advantageous strategy for the group; however, on an individual level, cheating is frequently more attractive. In a fishery, one can choose to cooperate by fishing only the regulated amount or not to cooperate, by fishing to maximize profits. Top-down management can help to emulate a cooperative result in fisheries, but it is costly and not always a viable alternative for development states. Here, we investigate elements of a fishing system that can be strategically managed to encourage a cooperative behaviour. Using bioeconomic data, we modelled an evolutionary game between two populations of fishers that differ if they cooperate or do not cooperate with a fishing restriction. We penalized players including risk tolerance and control perception, two social parameters that might favour cooperation. We assessed the degrees to which risk tolerance and control perception affect the cooperative behaviours of fishers in a restricted fishing effort small-scale fishery (RSSF) in southern Brazil. We also assessed the likelihood of a scenario wherein a cooperative strategy can evolve and dominate the system. We identified dominance and coexistence outcomes for the RSSF. Sensitivity analyses suggested that both control perception and risk tolerance could facilitate a cooperative outcome for the fishery.

## 1. Introduction

It is widely accepted that the 'tragedy of the commons' [1] or the 'tragedy of free for all fishing' [2] is the core of the issue in a Malthusian overfishing perspective [3]. Combined with technological advances, globalization of the fishing industry,

marginalization and inappropriate policies, such perspective can turn the open and unregulated access of common-pool resources unsustainable [3]. Any successful solution, whether applied through governmental intervention or through organization of a concerned community, will depend upon cooperation [4].

Despite potential costs to individuals, cooperation is 'good'—in terms of fitness—for the members of a group [5]. From the perspective of the group, the individual cost, if any, is rationally justified. However, for all individuals in a cooperative situation, there is a more attractive choice: cheating [6]. This paradox presents a dilemma for stakeholders in social-ecological systems: how to deal with cheaters? In fisheries, independent of the scale, this dilemma waylays fishers, decision-makers, researchers, and all of those concerned about building a sustainable way to fish [7,8]. Cooperative behaviour in this complex conflict, can be explored through evolutionary game theory to discern tendencies or produce predictions concerning each agent's choices [9].

Game theory is a mathematical approach to studying the strategic interaction of independent agents and their interests [10]. John Von Neumann and Oskar Morgenstern introduced the field to the scientific community in their studies on human behaviour in an economic context [11]. In biology, John Maynard Smith and George Price brought a game-theoretical framework to study evolution [12]. Evolutionary game theory considers populations of fixed strategies players interacting randomly with each other [13]. A population succeeds when it reproduces its strategy for future generations. Pay-offs for this game are interpreted as reproductive fitness.

In fisheries games, the pay-offs combine economic relationships drawn from fisheries (e.g. revenue, costs, market price) and ecological information about the stock (e.g. growth rate, stock size, catchability), resulting in so-called bioeconomic models [14]. Game-theoretic models for fisheries have focused on international waters, and applications of this framework on national/regional scale fisheries have yet to be sufficiently explored [9]. However, small-scale fisheries are crucial in the health of fishing ecosystems [15] albeit the present scarcity of data has direct consequences upon management strategies [16]. Managing strategies for fishers involve harvesting quotas, fishing effort restrictions, individual transferable quotas, protected areas and co-management [17–23], requiring cooperation from stakeholders both to implement and to maintain the system.

In a restricted fishing effort small-scale fishery (RSSF), there is regulation of the number of fishing units permitted to harvest. Social dilemmas appear when agents must decide whether to obey the restriction or to cheat and exceed the quota [24]. Key factors include both risk [25] and control [26] perceptions. A fisher's attitude towards risk (uncertainty) is an important issue to model decision making in fisheries [27]. Risk tolerance has been incorporated before in fisheries game theory models [28] and has being defined by theoretical and empirical studies as the ability to support uncertainty. In fisheries, it is influenced by the sense of availability of the resource in the future, fisheries prices' changes and environmental instabilities [25,28,29]. On another side, compliance, in criminal literature, is basically associated with certainty and severity of a punishment [30]. The control perception, in our study, is the amount of certainty that the fisher has about being punished [31]. Authorities' control over fisheries can be frequent or infrequent depending on the system, and the perception among fishers of control initiatives also factors into fishers' compliance [32].

We constructed a bioeconomic model for an RSSF using game theory. There is an extensive literature on the application of game theory to fisheries, and different approaches have been used in the last decades: two-players games, sequential games and coalitions games, to cite a few [33]. The seminal paper that analysed fisheries in a two-players game context was based on the classical Prisoner's Dilemma, which assumes that cheating is an equilibrium solution [34]—although non-cooperation does not automatically mean a negative outcome [33]. To remove the temptation to cheat, our model assumes *a priori* that both risk tolerance and control perception can interfere in fishers' behaviour. We sought to illuminate the interactions of social features with the bioeconomic aspects of this system, including the following: (i) are there combinations of values of risk tolerance and control perception that neutralize the system's natural tendency towards cheating dynamics and, in consequence, facilitate cooperation? and (ii) is there a scenario wherein cooperative strategies can evolve and dominate the fishery game? We set our model with parameters empirically adjusted to a case study focusing on an artisanal shrimp fyke net fishery in southern Brazil. In the last decades, game theory in fisheries has overwhelmingly focused on larger fleets and relatively simpler dynamics compared to artisanal fisheries close to open access, as the case here explored. We expect that our model helps to shed light on the conditions to the emergence of cooperation in small scale fisheries, which is absolutely essential in developing countries that do not have the capacity for top down enforcement and requires alternative and strategic management plans [35].

**Table 1.** List of biological, economic and social parameters used in the model.

| notation | description | unit |
| --- | --- | --- |
| $K$ | carrying capacity | Kg (biomass) |
| $B$ | stock size | Kg (biomass) |
| $r$ | growth rate | individual/time |
| $f$ | fishing effort | fishing units (nets, fishers, vessels, etc.) |
| $f^*$ | fishing effort regulated by law | fishing units (nets, fishers, vessels, etc.) |
| $q$ | catchability | percentage |
| $P$ | fish market price | Reais (Brazil's currency) |
| $c$ | fishing unit cost | Reais (Brazil's currency) |
| $\delta$ | behaviour penalty | percentage |
| $\alpha$ | fishers' control perception | percentage |
| $b$ | fishers' risk tolerance | percentage |
| $H$ | harvest | Kg (biomass) |

## 2. Methods

### 2.1. The model

Let $A$ and $B$ be two populations of fishers in a RSSF. Each population has a fixed strategy: to cooperate and to fish within the restrictions placed by a regulatory agency (cooperators), or not to cooperate and to fish pursuing individual profit maximization (non-cooperators or cheaters). In this game, we assume that the fish stock is composed of only one species in logistic growth. Catchability, a gear efficiency parameter, is assumed to be constant. Generations will be the fishing seasons, as stages in repeated games. Additional parameters used in the model are listed in table 1.

A behaviour penalty ($\delta$) is calculated using the risk profile ($b$), control perception ($\alpha$), and the growth rate of the stock ($r$). This penalty varies from zero, when cooperators have the maximum penalty, to one, when cheaters have the maximum penalization. We calculated this parameter by

$$\delta = \frac{\alpha}{1 + b \times r}. \tag{2.1}$$

The bottom expression shows the risk profile ($b$) and growth rate ($r$) in a negative relationship with the population penalty. High reposition of the stock (growth rate is high) will favour cheaters because stock will recover fast, reducing the sense of overharvesting and that a restriction is indeed necessary. Risk profiles vary from zero to one and show whether the fisher has a high ($b \gg 0$) or low ($b \gg 1$) risk tolerance. In this model, high risk tolerance will favour cooperators because, in this case, uncertainty will not drive fishers to fish more. Alternatively, behaviour penalty is directly proportional to control perception [30]. When perception of control is low (close to zero), for instance, the cheaters are favoured.

### 2.2. The stock dynamics

Gulland's harvest equation [36] shows the relationship of harvest to stock size: $H = F/(F + M) \cdot B \cdot (1 - e^{-(F+M) \cdot t})$. In time $t$, $H$ represents the amount harvested; $M$ and $F$ represents, respectively, stock mortalities by harvest and by natural causes and $B$ captures the biomass of the stock. We considered $t = 1$ because of the short fishing season. As natural mortality is a biological characteristic, we are assuming it to be constant and to not play an essential role in the dynamics of the system. For that reason, we assigned the value of natural mortality to $M = 0$. We assume $F = q \cdot f$ because fishery mortality depends on fishing effort and catchability. So $H = B \cdot (1 - e^{-q \cdot f})$. The harvest equation was used to calculate pay-off's matrix.

**Table 2.** Cooperators and non-cooperators encounters. ($\pi_{XY}$ is the pay-off for an individual from the population of $X$ in an encounter with an individual of population of $Y$. $C$ indicates cooperator and $N$ non-cooperator.)

| | cooperators | non-cooperators |
|---|---|---|
| cooperators | $\pi_{CC}$ | $\pi_{CN}$ |
| non-cooperators | $\pi_{NC}$ | $\pi_{NN}$ |

## 2.3. The game

We are playing a symmetric, simultaneous, deterministic, repeated evolutionary game. After setting populations' strategies, each one receives their pay-offs depending on the bioeconomic model. $\pi_{CC}$ is the pay-off for cooperators that meet each other. When a cooperator found a cheater, cooperators receive $\pi_{CN}$ and cheaters receive $\pi_{NC}$. Cheaters encounters with themselves has a pay-off measured by $\pi_{NN}$. The first pay-off index indicates the individual that is receiving the pay-off, either a cooperator or a non-cooperator (cheater). The other index indicates who meets with the pay-off receiver. $C$ stands for 'cooperator' and $N$ for 'non-cooperator'. Table 2 shows all possible encounters with the respective pay-offs.

Because risk tolerance and control perception intervene in the decisions of population $i$ we multiplied its pay-off by $\delta$ for cooperators or by $(1-\delta)$ for non-cooperators. For instance, $\pi'_{CN} = \delta \cdot \pi_{CN}$ and $\pi'_{NN} = (1-\delta) \cdot \pi_{NN}$. Individual pay-offs consist of all revenues minus all costs during the fishing season. The expected pay-off for each population (fitness) was calculated using the frequency of the population and the pay-offs for each individual meeting. Cooperators fish the resource respecting fishing effort restrictions. When a cooperator meets a cheater, she/he will fish with a regimented effort while the cheater will try to maximize her or his profit. When cheaters find each other, they will both try to maximize their profit at the same time, regardless of restrictions, generating rent dissipation owing to fishing race [37]. Pay-offs on each case are presented hereafter and detailed in the electronic supplementary material, S1:

$$\pi'_{CC} = \delta \cdot (B(1 - e^{-q \cdot f^*}) \cdot P - c \cdot f^*),$$

$$\pi'_{CN} = \delta \cdot \left( \frac{(c - B \cdot P \cdot q) \cdot f^*}{\ln(c/(B \cdot P \cdot q))} - c \cdot f^* \right),$$

$$\pi'_{NC} = (1 - \delta) \cdot \left( \frac{\ln(c/(B \cdot P \cdot q)) - q \cdot f^*}{\ln(c/(B \cdot P \cdot q))} B(1 - (c/(B \cdot P \cdot q)) \cdot P - c \cdot \left( -\frac{\ln(c/(B \cdot P \cdot q))}{q} - f^* \right) \right),$$

$$\pi'_{NN} = (1 - \delta) \cdot 0 = 0 .$$

## 2.4. Manipulating the model

We manipulated the model to produce findings independent of stock size ($B$). We divided all pay-offs by the carrying capacity ($K$) and introduced two new parameters. The parameter $B' = B/K$ measures the relative stock size and $c' = c/K$ measures the fishing cost of a unit of the stock when it reaches carrying capacity. Manipulated pay-offs can be accessed in the electronic supplementary material, S2.

## 2.5. Replicator's equation

To repeat the game over generations with a fixed pay-off matrix we used the replicator's equation:

$$\frac{dx_C}{dt} = x_C \cdot (1 - x_C) \cdot [\, \text{fit}_C(x) - \text{fit}_N(x)].$$

Here, $dx_C/dt$ is the variation in cooperators' relative frequency and $x = (x_C, x_N)$ is the population's composition with $x_C + x_N = 1$ within every time interval. $\text{fit}_C = x_A \cdot \pi_{CC} + x_B \cdot \pi_{CN}$ and $\text{fit}_N = x_A \cdot \pi_{NC} + x_B \cdot \pi_{NN}$ are fitness values for cooperative and non-cooperative individuals, respectively.

## 2.6. Solving the game

The outcomes for the game were: cooperative dominance, cooperative coexistence, non-cooperative coexistence, no fishing situation, bistability and non-cooperative dominance. Cooperative dominance

**Table 3.** Conditions for the outputs of strategic dynamics for the game. (Here, $\pi_{CC}$, $\pi_{NC}$, $\pi_{CN}$ and $\pi_{NN}$ are values taken from the pay-off matrix.)

| conditions | output |
| --- | --- |
| $\pi_{CC} > \pi_{NC}$ and $\pi_{CN} > \pi_{NN}$ | cooperative strategy dominates |
| $\pi_{CC} < \pi_{NC}$ and $\pi_{CN} < \pi_{NN}$ | non-cooperative strategy dominates |
| $\pi_{CC} > \pi_{NC}$ and $\pi_{CN} < \pi_{NN}$ | bistability |
| $\pi_{CC} < \pi_{NC}$ and $\pi_{CN} > \pi_{NN}$ | coexistence |

was defined as when a cooperative strategy was dominant upon non-cooperative one (Nash equilibrium). The same applies for non-cooperative dominance. Coexistence occurred when both strategies (populations) equilibrated the system with different frequencies. When the frequency of the cooperative population was higher, we defined it as cooperative coexistence. When the opposite occurred, we defined non-cooperative coexistence. No fishing situation occurred when fishing costs were higher than fishing revenue, making it unviable to fish. Bistability is a scenario where both populations could dominate. It has an instable coexistence point that relies between the dominance of each player. To determine outcome in each simulation we used conditions described in table 3. Algorithms are presented in the electronic supplementary material, S3. The equilibrium frequency in all coexistence outcomes was calculated, as shown in [13] by

$$\bar{x} = \frac{\pi'_{NN} - \pi'_{CN}}{\pi'_{CC} - \pi'_{CN} - \pi'_{NC} + \pi'_{NN}}.$$

## 2.7. Playing the game

At the beginning, the game represents the situation in which one population of cooperative individuals are coexisting with a population of non-cooperative individuals. $x_C$ and $x_N$ are the frequencies of the respective populations. Random encounters occur between these individuals and, for all of them, pay-offs for each individual are accounted. Using the frequencies, we calculate expected pay-offs for each population. These pay-offs represent the reproductive fitness and replicator's equation uses them to decide how would populations' frequencies be in the next generation. A population with better fitness will grow better, together with its behaviour strategy. As the game evolves, one population could dominate making the opponent's frequency null, or, the two populations could coexist with a stabilized frequency for each of them. Based on the game conditions we could predict what would happen over the generations.

## 2.8. The system

We fitted the model to RSSF in Laguna, southern Brazil. This fishery targets shrimp (*Farfantepenaeus paulensis* and *Farfantepenaeus brasiliensis*) in a lagoon complex using an unusual shrimp fyke net (used only in southern Brazil) since the decade of 1980 [38]. This passive fishing apparatus has two sleeves, a conic tunnel, and a bagger, and attracts fish using a lighted sign. Using a unique LED lamp, six fyke nets can be combined in a circular layout (electronic supplementary material, S4), trapping shrimp from all directions [39]. This fishery, and particularly this resource, is very important for the surrounding communities, both economically and culturally [38]. Local ordinance no. 32/1998 prescribes that only one location with six nets can be used for each licensed professional fisher [40]. The number of shrimp fyke nets found is more than the number of licensed fishers, indicating illegal harvesting [38]. Personal observations (E.Z.D. Azevedo, D.V. Dantas, F.G. Daura-Jorge 2018–2019) indicate that fishers indicated that the use of only one location restricts too much of the fishery, and relayed that without proper enforcement, some fishers use 70 or more locations. Concerned communities requested reviews of license regulation and for the approval of three fishing locations in a fishery agreement proposal, each spot with six nets. We applied our model to this system assuming the cooperation strategy, with fishers using only 18 nets, in accordance with the fishery agreement ($f^* = 18$). Some parameters were empirically determined using personal observations, technical reports, and investigations upon this fishery as shown in the electronic supplementary material, S5.

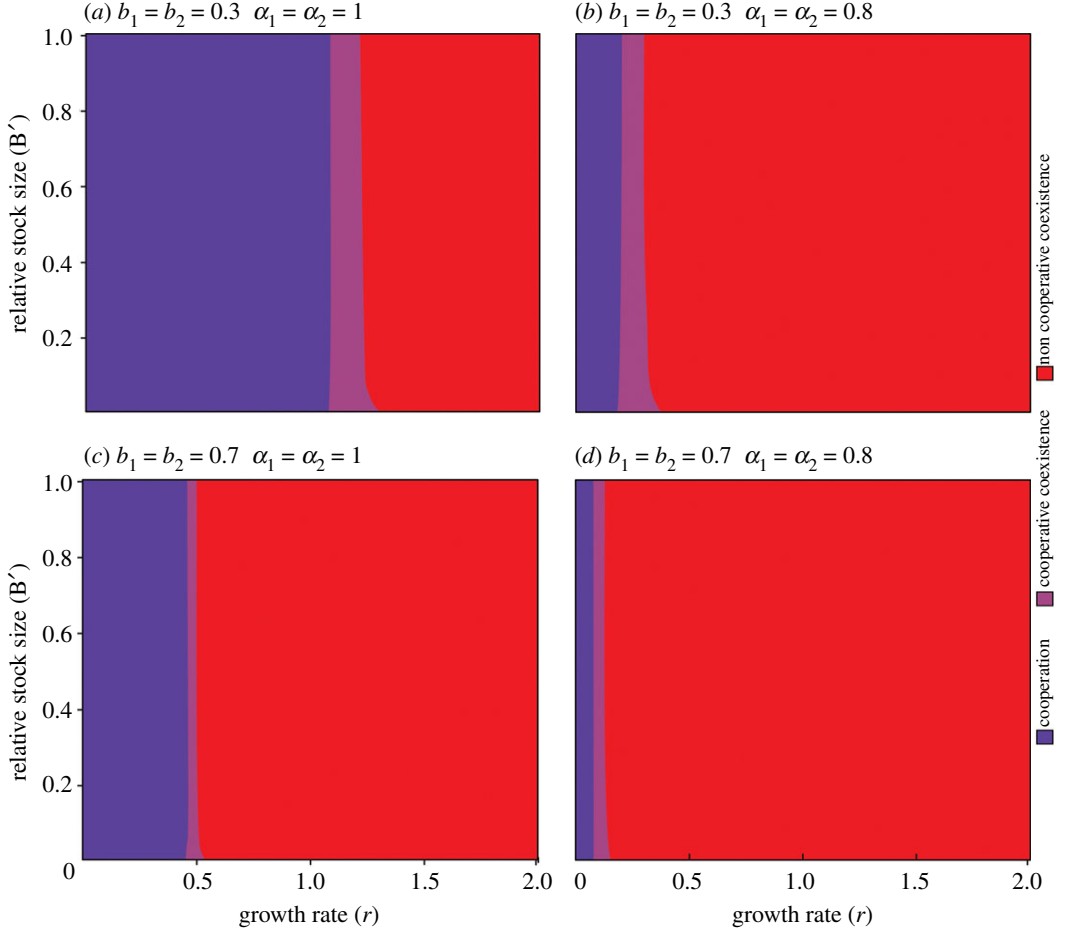

**Figure 1.** Cooperative dominance (blue), cooperative coexistence (pink) and non-cooperative coexistence (red) outcomes in two high risk tolerance ($b = 0.3$) scenarios (*a*) with a symmetric high perception of control ($\alpha = 1$) and (*b*) a symmetric low perception of control ($\alpha = 0.8$), and in two low risk tolerance ($b = 0.7$) scenarios (*c*) with a symmetric high perception of control ($\alpha = 1$) and (*d*) with a symmetric low perception of control ($\alpha = 0.8$). Every parameter beside $\alpha$ is the same in both scenarios ($b_1 = b_2 = 0.3$, $p = 23$, $q = 0.023$, $c' = (2 \times 10)^{-5}$).

## 2.9. Sensitivity analyses

In all simulations, we fixed the values for parameters $P$, $q$, $c'$ and $f^*$ ($P = 23$, $q = 0\ 0.023$, $c' = 2 \times 10^{-5}$ and $f^* = 18$). To build a scenario, we set values for $\alpha_1$, $\alpha_2$, $b_1$ and $b_2$ and the algorithm generated a game for each $B'$ and $r$ value from zero to one and between zero and two, respectively. In the coloured graphs, each colour identifies one of the possible outcomes. We set four different scenarios according to risk tolerance and control perception. For each of these scenarios, we analysed cooperative strategy frequency equilibria and how they evolve according to growth rate and to relative stock size. Finally, we present two scenarios in which cooperation invades a non-cooperative population as a strategy.

# 3. Results

## 3.1. Risk tolerance and control perception scenarios

When risk tolerance is high, there is an evident effect of control perception upon cooperative outcomes (figure 1*a,b*). Desirable outcomes for the system include both cooperative dominance and cooperative coexistence, in opposition to an undesirable non-cooperative coexistence. The most frequent outcome in figure 1*a* is the cooperative dominance with some cooperative coexistence for growth rate ($r$) values between 1.1 and 1.3. For $r > 1.3$, we found non-cooperative coexistence outcomes. Figure 1*b* shows a very small cooperative dominance range ($0 < r < 0.2$). Cooperative coexistence appeared in figure 1*b* when $0.2 < r < 0.33$. Non-cooperative coexistence appeared in figure 1*b* for almost every game with $r > 0.33$.

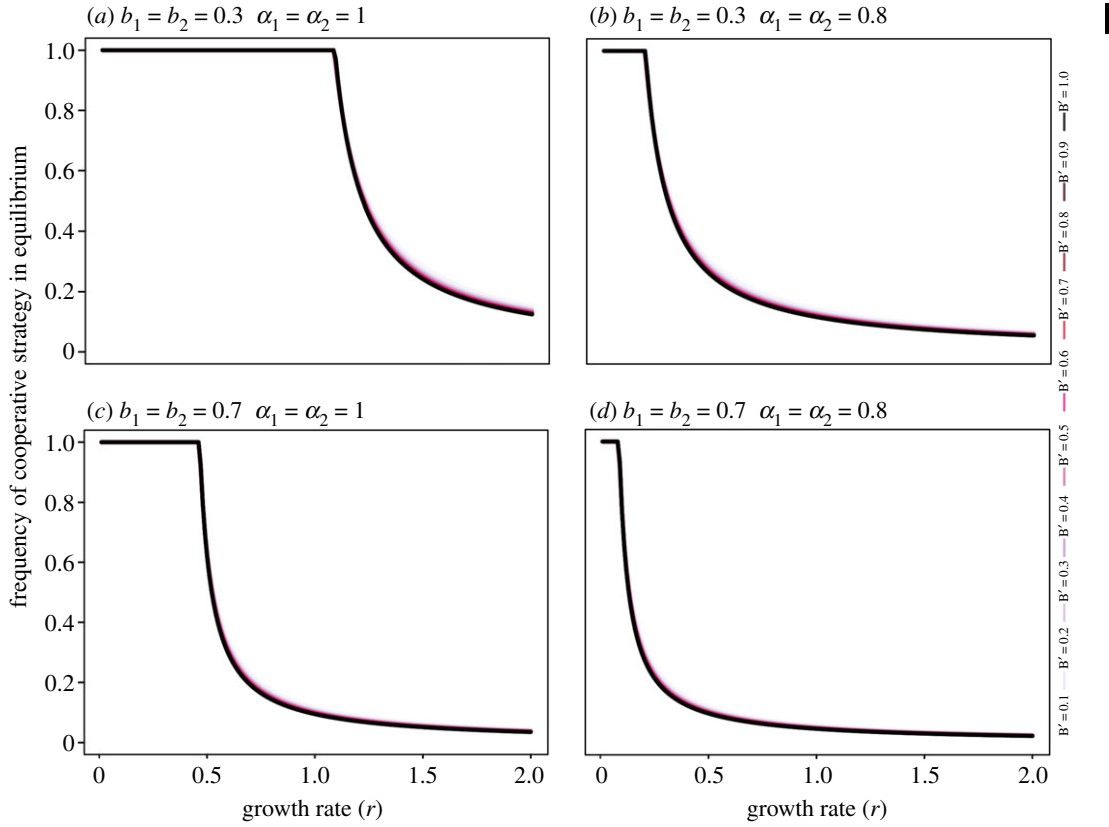

**Figure 2.** Interactions of cooperative strategy's frequency with growth rate in multiple scenarios: (*a*) with high risk tolerance and high perception of control, (*b*) with high risk tolerance and low perception of control, (*c*) with low risk tolerance and high perception of control, and (*d*) with low risk tolerance and low perception of control. Each coloured line represents a different relative stock size (B′) value. Besides risk tolerance and control perception, all parameters are constant ($p = 23$, $q = 0.023$, $c' = (2 \times 10)^{-5}$).

For low-risk tolerance scenarios, the perception of control also had a positive effect on cooperative outcomes (figure 1*c*,*d*). Cooperation strategy dominated when $0 \leq r \leq 0.46$ and when $r$ increased, the frequency of the cooperative strategy decreased. On the other hand, figure 1*d* shows primarily non-cooperative coexistence outcomes. Cooperative domination and cooperative coexistence take place only when $r < 0.14$. For all scenarios, high values of growth rate discouraged cooperation.

High-risk tolerance with high control perception facilitates cooperation (figure 1*a*) and low risk tolerance with low control perception discourages cooperative behaviour (figure 1*d*). The sensitivity of the model was higher for control perception than it was for risk tolerance.

## 3.2. Growth rate as driver for the cooperative behaviour

Growth rate ($r$) and risk tolerance ($b$) are components that control fishers' behaviour in the model (equation (2.1)). Relative stock size (B′), however, influenced outcomes indirectly, through the pay-off functions. Figure 2 shows that the growth rate is a better driver of the frequency of cooperative strategy than relative stock size. While variations in growth rate (horizontal axis) shape the curve, variations in B′ value (coloured scale) result in overlapping curves. When coexistence for both cooperative and non-cooperative strategies is the outcome for the game, the frequency of the former decreases as growth rate increases.

## 3.3. Invasion of the cooperative strategy

Figure 3 shows two different scenarios in which a population with 90% non-cooperative individuals (the residents) and 10% cooperative individuals (the invaders) is simulated. The first scenario (figure 3*a*) has high-risk tolerance and high control perception and encourages the invasion of cooperation by increasing the fitness of that strategy and, consequently, increasing its frequency dominance. By contrast, low-risk tolerance and low perception of control scenarios do not allow cooperative behaviour to invade

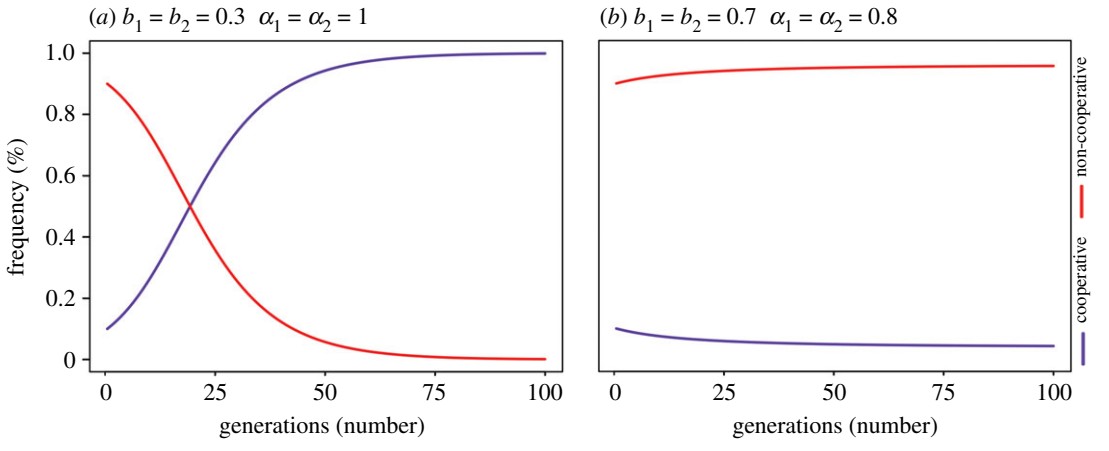

**Figure 3.** Dynamics of the strategic game between cooperative (blue) and non-cooperative (red) fishers in (*a*) high and (*b*) low risk tolerance and control perception scenarios using the replicator's equation. The initial population composition is 10% of cooperative individuals and 90% of non-cooperative individuals. Both scenarios have the same relative stock size (B′) and growth rate (*r*) (B′ = 0.3 and *r* = 1.0). Besides risk tolerance and control perception all parameters are constant ($p = 23$, $q = 0.023$, $c' = (2 \times 10)^{-5}$).

(figure 3*b*) and after multiple generations, the fitness of the cooperative strategy does not increase sufficiently to overcome non-cooperative behaviour and stabilizes in 4%.

## 4. Discussion

This study focused on the interactions of risk tolerance and control perception with the bioeconomic aspects of RSSF. If we remove control and risk effects from the model, non-cooperative coexistences result for all combinations of $r$ and $B'$ values (an all-red graph). This indicates that the game at its base state does not inspire cooperation. Manipulating risk and control effects in the fishery could be a way to obtain cooperative strategy dominant scenarios.

Using empirical information from Laguna's shrimp fyke net fishery, we were able to produce scenarios where cooperation was the most frequent outcome. The panorama presented in figure 1*a* is a 'good' scenario: high-risk tolerance and high control perception overcame the temptation to cheat and lead to a dominance of cooperation in most of the outcomes, indicating that cooperation could exist for some combination of these social parameter values. When control perception became low, either with high (figure 1*b*) or low (figure 1*d*) risk tolerance, the domination of cooperative strategies almost vanished as an outcome indicating a stronger influence through control perception when compared with risk tolerance. In the 'bad' scenario (figure 1*d*) we saw 80% of the fishers harvesting as much as possible to achieve maximum profit (figure 2*d*) while, on the other side of the water, fish stock significantly decreases owing to overexploitation, in a probable future collapse [41]. Here, fishers did not feel that restrictions were enforced sufficiently, and they would not tolerate the risk of stock uncertainty ($\alpha_1 = 0.8$ and $b = 0.7$). In a model with only risk tolerance and growth rate variables influencing the tendency to cooperate (an absence of control perception), analyses in [25] showed differences in cooperation behaviour from high risk tolerance scenarios (most of the outcomes were cooperative) to low-risk tolerance scenarios (few cooperative outcomes). The 'no fishing zone' appeared in all other scenarios presented in [25] but did not appear in either of ours.

Our results showed the growth rate of the stock ($r$) as a driver in the frequency of cooperative strategies (figure 2). As individuals sense the stock's characteristics, it might be important for management to focus on fishers' relation to the future of their resource. It is important to note that from the perspective of the fishers, the first result of changing from competition to cooperation is a reduction in revenue [42] and fishers' perceptions of the stock can differ from each other and also from authorities or experts in a shifted baseline [43]. Trisak's model also presented growth rate ($r$) as a driver for cooperative behaviour, and relative stock size encouraged non-cooperative behaviour in high risk scenarios [25].

In visualizing the invasion of cooperative strategies in a non-cooperative population, our results indicated that both risk tolerance and control perception are essential to guaranteeing the domination of cooperation. If the scenario is not good (figure 3*b*), the temptation to cheat will prevent cooperation

from arising in the system. On the other hand, if one could produce desirable values for risk tolerance and control perception (figure 3a), a cooperative strategy could invade and dominate the fishing system, producing a new equilibrium state for the game.

These conclusions raise a critical question: how could we strategically manipulate control perception and risk tolerance in real life? The simplistic answer is to control the fishery effectively and to guarantee that there will always be fish for everybody. Considering the difficulty of bringing these strategies to reality, especially in a developing state, we refer to examples of management in local fisheries. An Amazonian protection area of sustainable use called Mamirauá uses joint management to control their most economical fishery [44]. The Pirarucu's (*Arapaima gigas*) stock is monitored by both fishers and researchers, requiring active participation from local inhabitants. Fishers are organized in an association of producers and they fish with a compliance-based transferable quotas scheme [44]. The local community helps to protect the lakes against invasions and illegal fishing activities. This type of protection area permits local people to remain interacting with biodiversity (habitation and exploration), helping management to achieve sustainable use [45]. Through joint monitoring, local perception of fishing control could rise as fishers control and manage their fishery. In addition, this example indicates that stabilizing stock size could be a way to manage risk influence in the system as constant values of catch through time could facilitate fishers to perceive the real state of the stock.

Fishing teams in Yucatán, Mexico [29], transferable quotas in Dutch fisheries [46], reform on regulations rules in small Pacific islands [35] are other initiatives that improved cooperation in fisheries management. Of course, these measures do not guarantee success for all the fishery systems [47]. Stability of the cooperative system, also a complex issue, should be an early concern for management agents to guarantee the viability of cooperation [48]. For our case study in Laguna, based on our results, we recommend a package of initial measures to better perceive the fishery's particular characteristics. Both fishers and authorities aim to resolve legislation problems in a way to benefit economic and ecological interests about the fishery, but information about the system (scientific research and local knowledge of ecologic and social aspects) are not existent or transparent for all. Producing and organizing this information transparently could be a first step to achieve better regulation rules. Over time, transparency could also decrease cheating and empower the local community [49]. Information and transparency could also diminish fishers' uncertainty feeling and by consequence the risk effect on their behaviour, which favours cooperation. In addition, as aforementioned, effective top-down enforcement is costly [50,51] increasing the seeking for other compliance alternatives such as fisher's self-management and social norms (raised by empowerment of local fishers) [46]. Our model suggest that such alternatives can motivate cooperation if it increases the control perception by fishers.

It is important to note that small-scale fisheries are more complex than our model could have captured in this study, leaving some challenges for future work. Food security and poverty are important social drivers that account for a portion of model fishers' cooperative behaviour [52]. Other types of pay-offs that would not be evaluated directly by money terms (such as tradition, social status or religion) could also contribute to fishers' decisions [53]. An analysis at an ecosystem level could also bring important insights and capture more complexity even for one-species fisheries [54]. However, even simplifying the system, our model offers some directions for a strategic management. There is no way to build a sustainable fishing scenario besides working together with all stakeholders in an interdisciplinary task. The temptation to overfish in regular fisheries is sufficiently high that it induces fishers not to behave cooperatively. Top-down enforcement costs for developmental states claim for better alternatives. Strategically manipulating risk tolerance and control perception, then, could be very useful management tools to mitigate overexploitation problems in small-scale fisheries, facilitating cooperative decisions.

Data accessibility. This article has no additional data.

Authors' contributions. E.Z.D.A planned and coordinated the study, performed the formal analysis and modelling, wrote the original draft, reviewed the final draft; D.V.D discussed the results, reviewed the manuscript and helped to write the final version; F.G.D-J. helped with the conceptualization and coordination of the study, supervised the study, discussed the results and reviewed the final version. All authors read and approved the final manuscript.

Competing interests. We declare we have no competing interests.

Funding. No funding was received for this research.

Acknowledgements. We thank Andrea Freire, Asteroide Santana, Eduardo Giehl, Eduardo G. de Farias, Micheli C. Thomas, Natália Hanazaki, Pedro Pintassilgo, Rodrigo P. Medeiros and Sergio Floeter for all the suggestions. We also thank the two anonymous referees for comments and insights about our work.

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
