## [Reviewer comments · Royal Society Open Science]

Review History

RSOS-200621.R0 (Original submission)

Review form: Reviewer 1

Is the manuscript scientifically sound in its present form?

Yes

Are the interpretations and conclusions justified by the results?

Yes

Is the language acceptable?

Yes

Do you have any ethical concerns with this paper?

No

Have you any concerns about statistical analyses in this paper?

No

Recommendation?

Accept as is

Comments to the Author(s)

Nice revision.

Decision letter (RSOS-200621.R0)

Dear Mr Zettermann Dias de Azevedo:

It is a pleasure to accept your manuscript entitled "Risk tolerance and control perception in a game-theoretic bioeconomic model for small-scale fisheries" in its current form for publication in Royal Society Open Science. The comments of the reviewer(s) who reviewed your manuscript are included at the foot of this letter.

on behalf of Dr Punidan Jeyasingh (Associate Editor)

Associate Editor Dr Punidan Jeyasingh Comments to Author:

Associate Editor: 1

Comments to the Author:

I thank the authors for a thorough revision of the ms based on comments from reviewers at ProcB. The revised version was reassessed by one (more critical) of the original reviewers. This reviewer is highly satisfied with the revisions. I agree with the expert - this version reads much better! I am happy to recommend the manuscript for publication.

Reviewer(s)' Comments to Author:

Reviewer: 1

Comments to the Author(s)

Nice revision.
